# Research on optimization of perforation parameters for formation fractures based on response surface optimization method

Wei Liu[1,2], Suling Wang[1,2]*, Kangxing Dong[1,2], Tiancai Cheng[1,2]

**1** School of Mechanics Science & Engineering, Northeast Petroleum University, Daqing, Heilongjiang, China, **2** Heilongjiang Key Laboratory of Petroleum and Petrochemical Multiphase Treatment and Pollution Prevention, Daqing, Heilongjiang, China

* 925950943@qq.com

**Data Availability Statement:** All relevant data are within the manuscript and its Supporting Information files

## Abstract

For staged multi-cluster fracturing, methods for controlling perforation friction to adjust the flow distribution of each cluster can effectively promote the uniform extension of multiple fractures but lacks a fast and quantitative optimization method for different perforation parameters of each cluster. By establishing a numerical model of single-stage three-cluster flow-limited fracturing under stress-seepage coupling, and based on the response surface optimization method, fully considering the impact of perforation parameters interaction among three perforation clusters, according to the regression equation fitted under the global response, the rapid optimization of perforation parameters of segmented multi-cluster fracturing model is realized. The results show that: in determining the three factors of the study, it is found that there is an obvious interaction between the number of intermediate cluster perforations and the number of cluster perforations on both sides, the number of cluster perforations on both sides and the diameter of intermediate cluster perforations, the response surface optimization method gives the optimal perforation parameter combination of three clusters of fractures under global response; When the perforation parameters were combined before optimization, the fracture length difference was 32.550m, and the intermediate perforation cluster evolved into invalid perforation cluster, when the perforation parameters were combined after optimization, the fracture length difference was 0.528m, the three perforation clusters spread uniformly, and there are no invalid clusters. At the same time, the regression equation under the response is optimized before and after the comparison between the predicted value of the equation and the actual simulation value. It is found that the estimated deviation rate of the equation before optimization is 1.2%, and the estimated deviation rate after optimization is 0.4%. The estimated deviation rates are all less, and the response regression equation based on the response surface optimization method can quickly optimize the perforation parameters. The response surface optimization method is suitable for the multi parameter optimization research of formation fracturing which is often affected by many geological and engineering factors. Combining with the engineering practice and integrating more factors to optimize the hydraulic fracturing parameters, it is of great significance to improve the success rate of hydraulic fracturing application.

**Funding:** Wang Suling host the Science Center Project of National Natural Science Foundation of China/Basic Science Center Project [Grant number 72088101]; Wang Suling host the Science and Technology Cooperation Project of Heilongjiang Province Science and Technology Plan [Grant number YS19A04]; Dong Kangxing host the China Postdoctoral Science Foundation [Grant number 2019M661249]; Dong Kangxing host the Postdoctoral Foundation of Heilongjiang Province [Grant number LBH-Z19123].

**Competing interests:** The authors have declared that no competing interests exist.

# Introduction

In recent years, the energy world has set off an "unconventional oil and gas resource revolution". The development of unconventional oil and gas resources has increased year by year, which is affecting the world's energy supply and demand pattern [1–3]. After multi-cluster perforation in each fracturing stage, multi-cluster fracture initiation and extension can be accomplished by pumping at one time, effectively reducing the construction cost for the exploitation of unconventional oil and gas resources and becoming one of the core technologies for the exploitation of unconventional oil and gas resources [4]. Production test data and literature show that for perforation clusters after staged multi-cluster fracturing, a small part of perforation clusters contribute to productivity by initiation and expansion, while a considerable part of perforation clusters fail to initiate and expand and thus become ineffective perforation clusters [5, 6]. In view of the problem that partial perforation clusters in segmented multi-cluster fracturing become invalid perforation clusters, scholars at home and abroad are constantly studying methods that can promote the uniform development of perforation clusters in horizontal well sections. Peirce [7], Bunger [8], Potapenko [9] and Lecerf [10] found that the spacing of perforation clusters will have the effect of stress interference on the development of fractures in the middle clusters. The method of optimizing the spacing of perforation clusters or non-uniform distribution of perforations is proposed to reduce the stress interference and realize the uniform extension of multiple perforation clusters. Wu [11] and Lecampion [12] stated that stress interference and dynamic flow distribution are two main factors influencing the balanced propagation of multi-cluster fractures. A single-stage three-cluster fracturing model was established, proving that the flow distribution could be effectively regulated by controlling the hole friction and the impact of stress interference on the non-uniform fracture propagation could be reduced. Zhao [13] and Li [14] established a segmental multi-cluster model and numerically simulated the number of perforation holes in different perforation clusters, so as to control the hole friction and achieve uniform distribution of the flow of each perforation cluster. However, the above methods only consider the impact of a single factor on the fracture propagation of the formation in the process of studying the staged multi-cluster fracturing model, and do not consider multiple factors and the interaction between multiple factors at the same time. Therefore, the research process of staged multi-cluster fracturing lacks an optimization method that can simultaneously consider and judge whether there is an interaction between multiple factors, and gives the optimal parameter combination under the global response.

Response surface optimization (RSM) has the advantage of fully considering the interaction among a large number of factors and quickly matching the optimal parameter combination among multiple factors under the global response based on the multiple linear regression model in mathematical and statistical methods. Zhu [15] applied the response surface optimization method in the process of sugar extraction from mulberry leaves; Li [16] applied the response surface optimization method in the tobacco baking process; Liu [17] applied the response surface optimization method in the vehicle multi-objective optimization; Zhang [18] applied the response surface optimization method in the structure optimization of cyclone separator. But, the combination of response surface optimization method and formation hydraulic fracturing has not yet been discovered. The staged multi-cluster fracturing of horizontal wells is often affected by many factors. Therefore, this paper proposes a method based on response surface optimization to optimize the perforation parameters of the staged multi-cluster fracturing model so that the fracturing model can be effectively control the perforation friction of each cluster, adjust the flow distribution of each cluster, and promote the uniform extension of multiple cracks.

## Methods

### Cohesive zone method

Cohesive Zone Method models are widely used for hydraulic fracturing fracture propagation simulation [19, 20]. The traditional linear elastic fracture mechanics are often singular in the process of fracture tip development, and the intensity of the singularity will bring numerical difficulties to the analysis and calculation. In the process of fracture propagation description, a cohesive zone method model can be characterized by the cohesive force zone and traction separation criterion. This effectively avoids the problem of singularity stresses at the crack tip in the calculation of traditional linear elastic fracture mechanics, thus eliminating the complicated process of calculating the stress intensity factor at the crack tip. The traction separation criterion of the Cohesive unit is shown in Fig 1.

The cohesion zone and traction separation criteria described by Dugdale [21] and Barenbaltt [22] are determined by the peak strength and fracture energy of the nominal stress-displacement curve. Before the Cohesive element reaches the damage, the linear elastic relationship is satisfied and then the damage and evolution process occurs [23]. At the initial

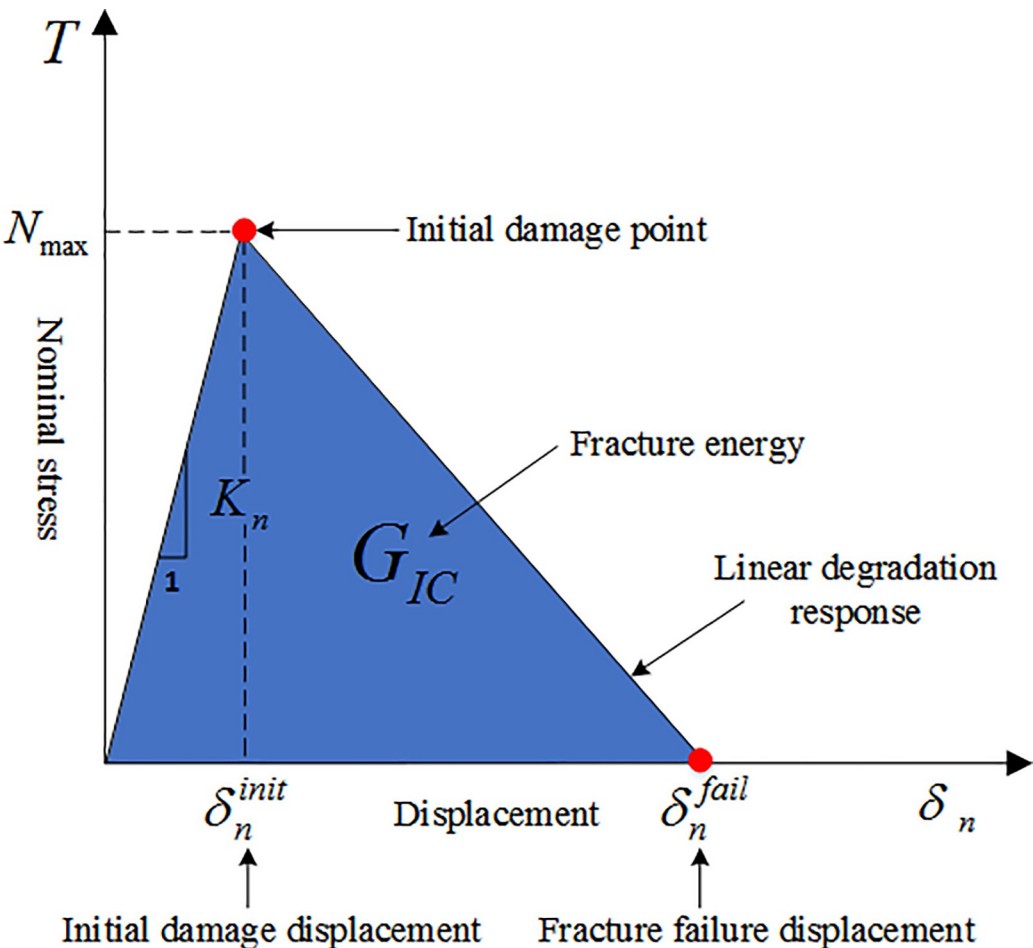

**Fig 1. Cohesive unit traction separation criterion.**

stage, the linear elastic constitutive calculation equation is given by:

$$t = \begin{bmatrix} t_n \\ t_s \\ t_t \end{bmatrix} = \begin{bmatrix} K_{nn} & 0 & 0 \\ 0 & K_{ss} & 0 \\ 0 & 0 & K_{tt} \end{bmatrix} \begin{bmatrix} \delta_n \\ \delta_s \\ \delta_t \end{bmatrix} = K\delta \tag{1}$$

Where $t$ represents traction, n, s, t indicates different directions; $K$ represents stiffness matrix; $\delta$ represents the displacement.

**Damage initiation criterion.** In the process of crack initiation and propagation, both tensile and shear stresses exist on the upper and lower surfaces of the element. Therefore, the quadratic nominal stress criterion is selected for the crack initiation to judge the initial damage; that is, the initial damage starts when the sum of the square of the stresses borne by the three traction forces and the critical stress ratio is 1. The equation is written as following:

$$f = \left\{ \frac{<t_n>}{t_n^0} \right\}^2 + \left\{ \frac{<t_s>}{t_s^0} \right\}^2 + \left\{ \frac{<t_t>}{t_t^0} \right\}^2 \tag{2}$$

**Damage evolution criterion.** Once the Cohesive unit reaches the damage initiation standard, it will enter the damage evolution stage. The scalar damage D is used to represent the overall damage of the crack. The initial value of the scalar damage D is 0. After the damage evolution model is defined, the value of D at the time of complete damage is 1 after the initial loading of the damage, which is the process of crack formation. The calculation equation is:

$$t_n = \begin{cases} (1-D)\bar{t}_n, \bar{t}_n \geq 0 \\ \bar{t}_n, \textbf{other} \end{cases} \tag{3}$$

$$t_s = (1-D)\bar{t}_s \tag{4}$$

$$t_t = (1-D)\bar{t}_t \tag{5}$$

$$D = \frac{\delta_m^f(\delta_m^{\max} - \delta_m^0)}{\delta_m^{\max}(f - \delta_m^0)} \tag{6}$$

Where $\delta_m^f$ represents effective displacement at complete failure, $\delta_m^0$ represents effective displacement at initial damage, D stands for total damage.

**Fluid flow properties in the damage zones.** After the damage and breakdown of the cohesive zone method model, two pathways can be taken. On the one hand, the fracturing fluid flows in the fracture plane, affecting the change of stress and strain on the fracture plane. On the other hand, fluid flow and exchange occur in the pores of the rock, affecting the change of stress and strain on the reservoir matrix. Among them, the flow of fluid in the fracture is divided into tangential flow and normal filtration. Assuming that the fluid is an in-compressible Newtonian fluid, the calculation equation of tangential flow is given as follows:

$$q = -\frac{w^3}{12\mu} \nabla p \tag{7}$$

$$\begin{cases} q_t = c_t(p_f - p_t) \\ q_b = c_b(p_f - p_b) \end{cases} \tag{8}$$

Where $q$ represents tangential flow rate, $\nabla p$ represents fluid pressure gradient in the fracture, $\mu$ represents fracturing fluid viscosity, $c$ represents filtration coefficient, $t$, $b$ represents upper and lower surfaces.

## Pipe flow unit and connection unit

In the process of staged multi-cluster fracturing in horizontal wells, the fracturing fluid is injected initially from the wellhead, flows through the casing and reaches each cluster. Due to the different frictions of each cluster, the fracturing is dynamically distributed among each perforation cluster. In the process of dynamic distribution, the hole flow distribution is uneven and the perforation cluster with little or no flow distribution is derived into an invalid perforation cluster. At present, some studies still make assumptions on the quantitative conditions of the flow rate of each perforation cluster. For example, the use of numerical simulation to realize the staged multi-cluster fracturing of horizontal wells and the real dynamic distribution process of fracturing fluid will be more realistic. Two units are introduced in this part: (1) the pipe flow unit, which is used to simulate the dynamic injection process of the fracturing fluid from the wellhead to each cluster; (2) the connection unit, which is used to simulate the perforating hole pressure drop and realize the dynamic flow distribution process of each perforation cluster.

**Pipe flow unit.**   The pipe flow unit simulation takes into account both the viscosity and gravity loss of the fluid in the pipe. Based on the Bernoulli equation in which the node height Z and length L of the pipe in the loss coefficient considers the loss along the pipe, the flow of single-phase incompressible fluid in the pipe section can be simulated. The equation for the pressure drop loss from the wellhead to each cluster perforation hole is given by:

$$\Delta p - \rho g \Delta z = (C_L + K_i)\frac{\rho v^2}{2} \tag{9}$$

$$C_L = \frac{fL}{D_h} \tag{10}$$

Where $p$ represents pressure, $z$ represents height, $v$ represents velocity, $\rho$ represents density, $g$ represents acceleration of gravity, $f$ represents friction factor, $L$ represents pipe length, $D_h$ represents pipe diameter.

**Connection unit.**   During staged multi-cluster perforation and fracturing of horizontal wells, when a large amount of fracturing fluid enters the formation through the perforation, the perforation acts similar to a throttle valve causing a certain pressure loss. The connection unit is the key to control the hole friction and adjust the flow distribution of each cluster. Different from the pipe flow unit, the connection unit ignores the length of the unit to avoid the loss along the passage caused by the length. As shown in Fig 2, the connection unit is composed of two nodes, where only the degree of freedom of pore pressure exists. Fluid flows in at point 1 and exits at point 2. The calculation equation of the perforation pressure drop is given

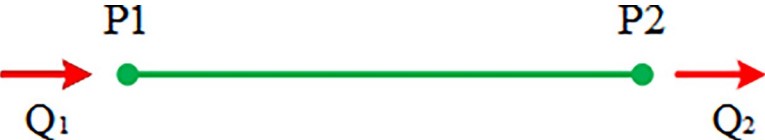

**Fig 2. Connection unit (Revised from reference [25]).**

by [24, 25]:

$$\Delta P_{fric}^{I} = p_1 - p_2 = \phi_p Q_I{}^2 \qquad (11)$$

$$\phi_p = 0.807249 \frac{\rho}{N^2 D_p{}^4 C^2} \qquad (12)$$

Where $\Delta P_{fric}^{I}$ represents perforation cluster friction, $I = 1 \sim n$ represents perforation cluster number, $N$ represents number of perforations, $D_p$ represents perforation diameter, $C$ represents wear coefficient.

## Unit and simulation method validation

In order to verify the effectiveness of pipe flow and the connection unit as well as prove the accuracy of the numerical simulation method, a three-dimensional three-cluster fracturing model similar to reference [11] was established as shown in Fig 3. The mesh generation of the three-dimensional fracturing structure model was shown in Fig 4. The wellborn and perforation hole were respectively set as the pipe flow and connection unit to test whether the flow of each perforation cluster was dynamically distributed. The simulation results were compared and analysed with the results of dynamic flow distribution realized in the references as shown in Fig 5. As can be seen, the simulation results of the setting pipe flow and connection units in this paper are in good agreement with the curves of the flow dynamic distribution realized in the literature. Under the same parameters of the three cluster perforations, the flow distribution is more than that of the 1 and 3 cluster perforations on both sides, and the flow distribution of the intermediate 2 cluster perforations is very less (approaching 0). The dynamic distribution process of downhole flow was simulated by using pipe flow unit and connection unit, which was in good agreement with references, thus proving the unit and simulation

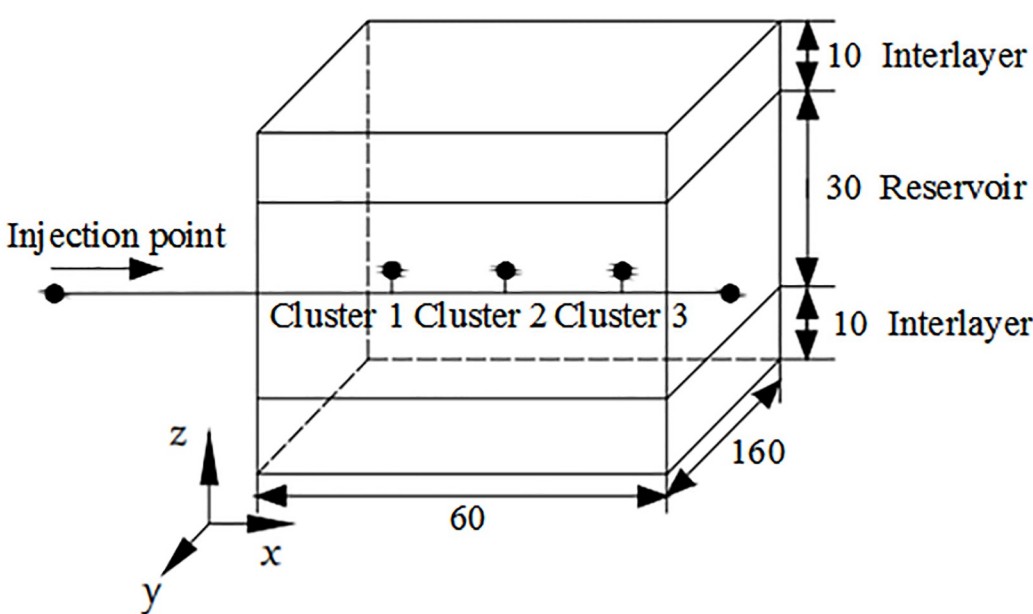

**Fig 3. Three-dimensional fracture structure model.**

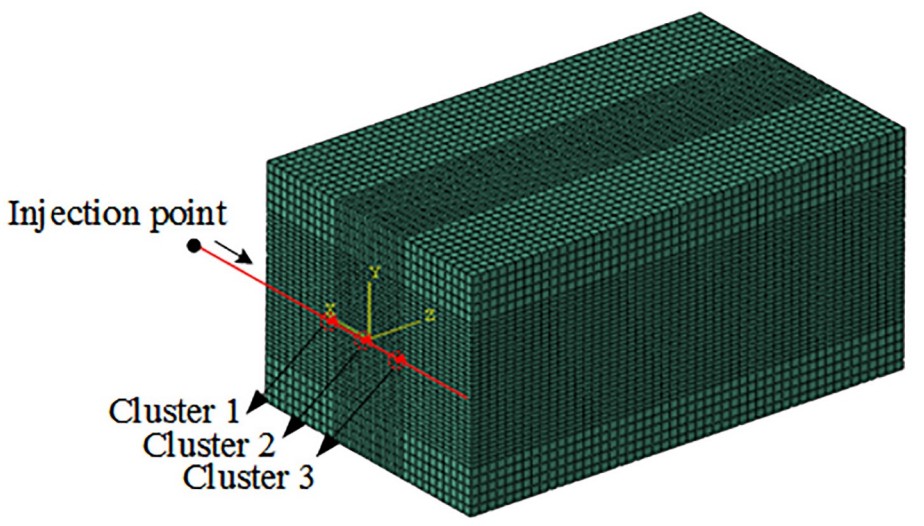

**Fig 4. Mesh of three-dimensional fracture structure model.**

**Fig 5. Comparison of simulation with reference results.**

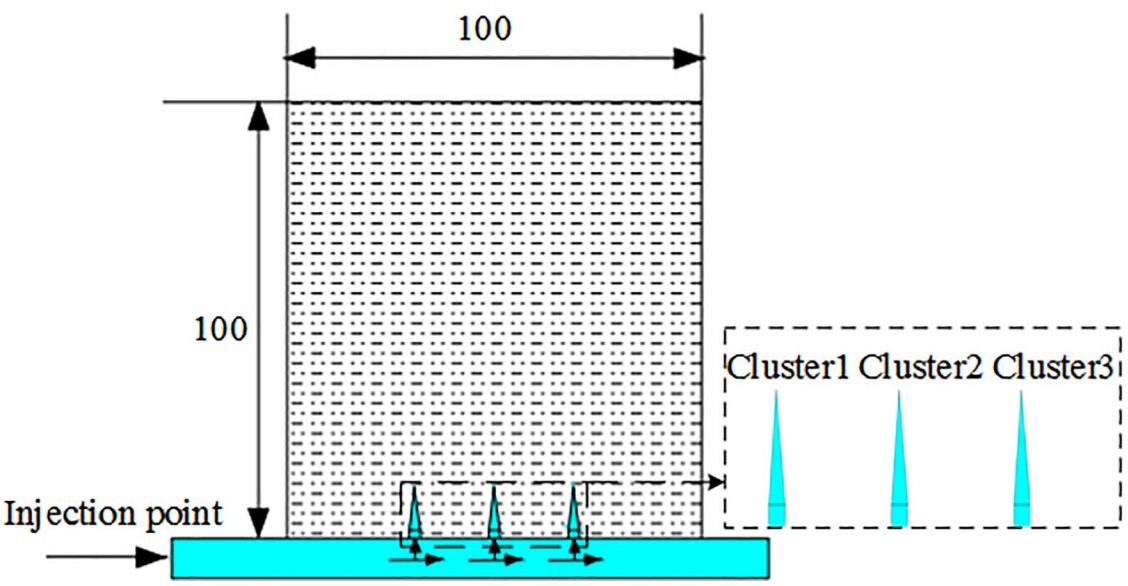

**Fig 6. Two-dimensional calculation model.**

method is correct. The above methods will be used to simulate the multi-cluster fracturing process of horizontal wells in the future.

## Calculation model and materials

Considering the calculation time and content of the three-dimensional model and the two-dimensional model, the two-dimensional model was selected to simulate the single-stage three-cluster fracturing process. By optimizing the perforation parameters of each cluster, perforation friction is controlled to adjust the uniform flow distribution of each cluster. With the three clusters of cracks evenly extended, the two-dimensional calculation model is shown in Fig 6. In order to avoid the impact of boundary conditions on the fracture length propagation of clusters, the model length and width are both set to 100m, and three perforation clusters are set. Some reservoir and perforation parameters of the two-dimensional calculation model are shown in Table 1.

Table 1. Calculation model materials parameter table.

| Parameter | Numerical value |
|---|---|
| Young's modulus | 20 GPa |
| Poisson's ratio | 0.2 |
| Fracture toughness | 0.56 MPa·m$^{0.5}$ |
| Tensile strength | 6 MPa |
| Maximum horizontal stress | 55 MPa |
| Minimum horizontal stress | 50 MPa |
| Number of Perforations | 12 |
| Diameter of perforations | 12 mm |
| Fracturing fluid viscosity | 10 mPa·s |
| Fracturing fluid density | 1010 kg·m$^{-3}$ |

## Response optimization design method

At present, control variable method or orthogonal test method is mainly used in the optimization design process of hydraulic fracturing crack propagation. The above two methods can only limit the design process of each factor to a given level and cannot conduct global optimization for a certain range of parameters. Therefore, the interaction between multiple factors cannot be taken into account, and the optimal design parameters obtained are often not the optimal parameter combination among multiple factors [26–28]. The response surface optimization design method enables horizontal optimization analysis to be continuously carried out for multiple influencing factors. It can overcome the defects of the control variable method and the orthogonal test method, which can only optimize the design and analysis of each isolated point [29, 30]. At the same time, within a certain range, it has the advantages of fewer test time, an accurate fitting equation, good prediction performance and can fully consider the interaction between different factors.

## Principle of the response surface method

Response surface optimization (RSM) is based on multiple linear regression models in mathematical and statistical methods and approximates the functional relations of implicit limit states by establishing polynomials of different orders. The expression between the system response evaluation index Y and the design factor variable x in the response surface design is as follows:

$$Y = \tilde{y}(x) + \delta \tag{13}$$

Where $\tilde{y}(x)$ represents the approximate function of the unknown function, $\delta$ represents total error.

Among them, if the Quadratic Response Surface Test Box-Behnken Design (BBD) and Central Composite Design (CCD) design methods are used to approximate the relationship between the system design variables and response indicators, a second-order calculation model is required to approximate the response surface [31].

$$\tilde{y}(x) = \beta_0 + \sum_{i=1}^{k} \beta_i \chi_i + \sum_{i=1}^{k} \beta_i \chi_i^2 + \sum_{i=1}^{k} \beta_{ij} \chi_i \chi_j + \varepsilon \tag{14}$$

Where $\beta_i, \beta_{ii}, \beta_{ij}$ represents odd function, $\chi_i, \chi_j$ represents basis function.

The process of response optimization for design factors using the response surface optimization method is shown in Fig 7.

## Response surface optimization design scheme

According to the perforation pressure drop calculation Eq (12), it can be seen that the number of perforating holes N and the diameter of perforating holes $D_p$ are the key factors affecting the frictional resistance of perforating holes that realize the flow distribution adjustment. At the same time, reference [14] found in the simulation process of the staged multi-cluster fracturing model that the calculation results of perforation clusters on both sides are symmetric and the number of cluster perforations on both sides can be set the same. Finally, three parameters are selected to optimize, respectively, the number of intermediate cluster perforations(A), the number of cluster perforations on both sides(B), and the diameter of intermediate cluster perforations(C). The optimal combination of parameters in the range tries to attain control of hole friction, adjust the flow distribution of each cluster, and effectively promote the uniform extension of multiple cracks.

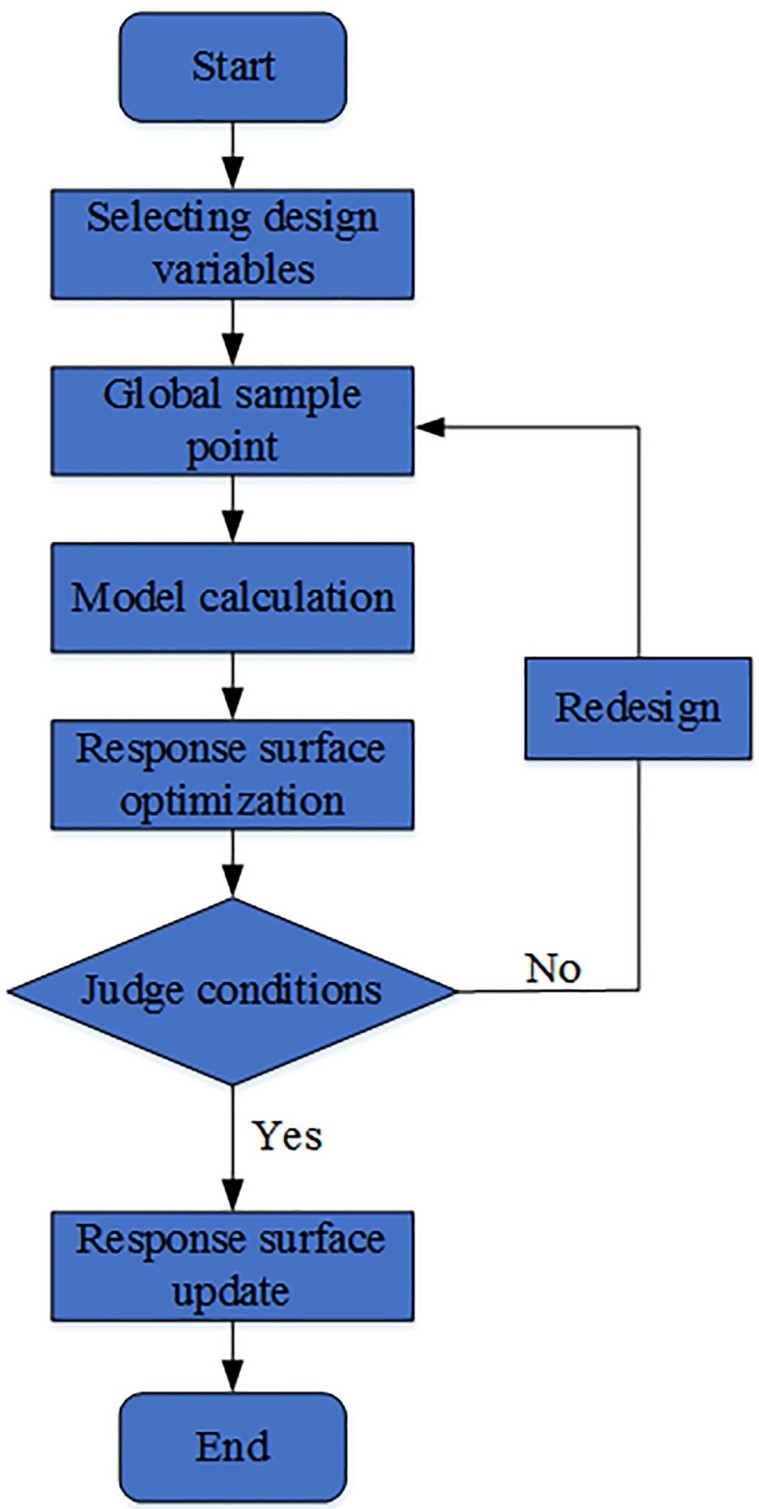

**Fig 7. Response surface optimization method process.**

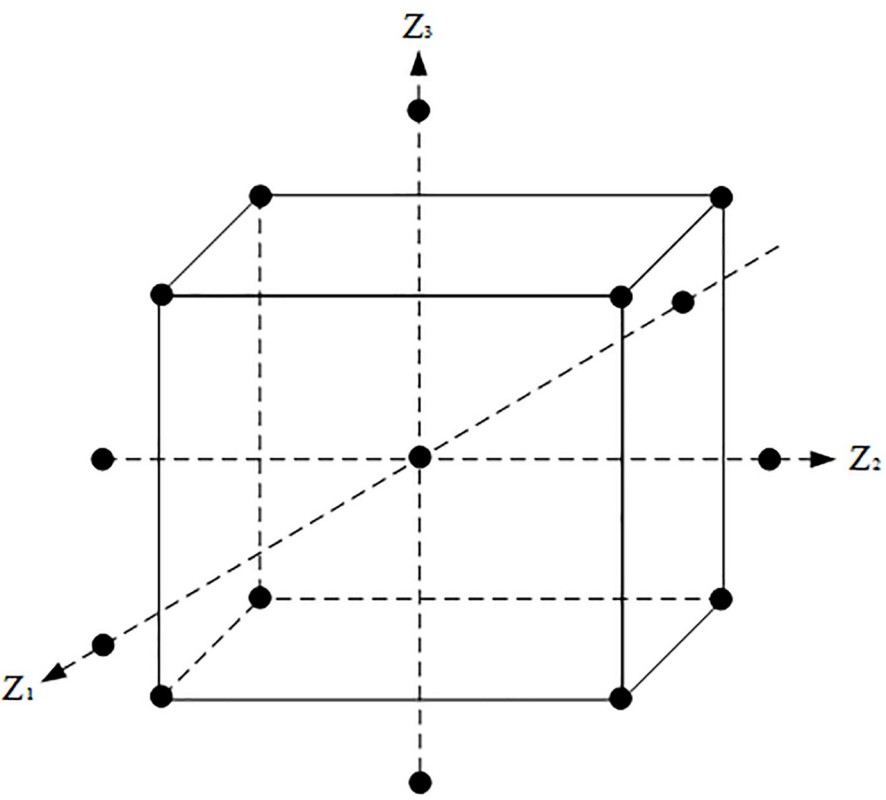

**Fig 8. CCD test design method.**

Response surface optimization experimental design methods include many forms, among which the more commonly used are Central Composite Design (CCD) and Box-Behnken Design (BBD). The spatial point distribution of CCD and BBD experimental design was shown in Figs 8 and 9. From the design points of CCD shown in Fig 8, it can be seen that the CCD design method have data beyond the original level. The results of BBD design points shown in Fig 9 are all within the set level range. In comparison, the BBD test design is more conducive to the design of perforation parameter optimization test points, and the CCD test design method has the inapplicability of exceeding the level of harm or violating the requirements of actual working conditions. Therefore, the paper selects the BBD design method to optimize the response of design factors. In the BBD design method, each factor takes 3 levels and is coded with (-1, 0, 1). The values of -1 and 1 are the low and high values, respectively, corresponding to the cube points, while 0 is the center point which is used to match the response surface design scheme and the result value. The horizontal interval values of the three optimization factors are shown in Table 2.

According to the interval value of the factor parameter range, 17 parameter combination schemes are given based on the experimental design software of the response surface optimization method, and the fracture length difference of the single segment three cluster two-dimensional fracture propagation model is numerically simulated and calculated. The sorting results are provided in Table 3.

## Model construction and test

For the response design scheme and results in Table 3 provided above, the second-order polynomial model is used to test its significance. According to the significance test results of the

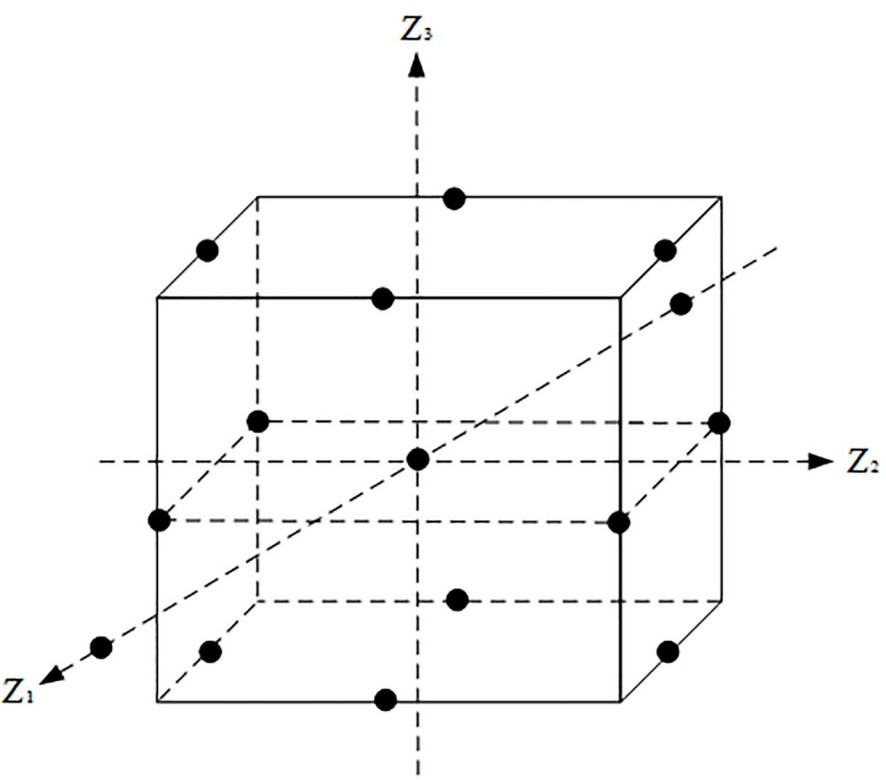

**Fig 9. BBD test design method.**

model, the arrangement is shown in Table 4. According to the results in Table 4, the P value is 0.0018, which is less than 0.01 indicating that the quadratic model adopted is significant. At the same time, according to the method of judging the significance, it is found that the number of intermediate cluster perforations(A), the number of cluster perforations on both sides(B), and the diameter of intermediate cluster perforations(C) have a significant impact on the slit length of the perforation cluster. Among them, significant interactions are observed between the number of intermediate cluster perforations(A) and the number of cluster perforations on both sides(B), the number of cluster perforations on both sides(B) and the diameter of intermediate cluster perforations(C). There is no interaction between the number of intermediate cluster perforations(A) and the diameter of intermediate cluster perforations(C).

According to the experimental design value, the multiple regression equation between the seam length difference and within the number and diameter of perforations is obtained. Through the expression, the global optimization can be carried out in the horizontal interval of the factors, and the optimal combination of parameters among the three parameters in the

**Table 2. Design factors and levels.**

| Options | Symbol | Level | | |
|---|---|---|---|---|
| | | -1 | 0 | 1 |
| Number of intermediate cluster perforations | A | 4 | 12 | 20 |
| Number of cluster perforations on both sides | B | 4 | 12 | 20 |
| Diameter of intermediate cluster perforations/mm | C | 6 | 12 | 18 |

**Table 3. Response surface design plan and results.**

| Number | Number of intermediate cluster perforations(A) | Number of cluster perforations on both sides(B) | Diameter of intermediate cluster perforations(C)/mm | Seam length difference/m |
|---|---|---|---|---|
| 1 | 0.000 | 1.000 | 1.000 | 30.3537 |
| 2 | -1.000 | 0.000 | 1.000 | 32.9299 |
| 3 | 0.000 | -1.000 | 1.000 | -23.7666 |
| 4 | 1.000 | 0.000 | 1.000 | 17.4154 |
| 5 | 0.000 | 0.000 | 0.000 | 32.5474 |
| 6 | 0.000 | 0.000 | 0.000 | 32.5474 |
| 7 | -1.000 | 1.000 | 0.000 | 52.8301 |
| 8 | -1.000 | -1.000 | 0.000 | 30.3537 |
| 9 | 0.000 | 0.000 | 0.000 | 32.5474 |
| 10 | 0.000 | 0.000 | 0.000 | 32.5474 |
| 11 | 1.000 | -1.000 | 0.000 | -22.0436 |
| 12 | 1.000 | 1.000 | 0.000 | 43.9413 |
| 13 | 0.000 | 0.000 | 0.000 | 34.5474 |
| 14 | 1.000 | 0.000 | -1.000 | 39.4493 |
| 15 | -1.000 | 0.000 | -1.000 | 43.1271 |
| 16 | 0.000 | 1.000 | -1.000 | 42.8325 |
| 17 | 0.000 | -1.000 | -1.000 | 29.9902 |

range can be quickly obtained.

$$y = 62.77 - 3.81A + 1.57B - 1.85C + 0.17AB - 0.06AC + 0.22BC$$
$$+0.05A^2 - 0.16B^2 - 0.09C^2$$

(15)

The value of the multivariate phase relation can reflect the accuracy of the fitting equation. If the correlation coefficient R-squared is approaching 1, it indicates that the response has a strong correlation. As a result of the experimental design scheme, the fitting correlation coefficient R-squared is 0.94, which is close to 1 in height, proving the accuracy of the fitting

**Table 4. Analysis of variance of test results.**

| Options | Sum of squares | Degrees of freedom | Mean square | F value | P value | Remarks |
|---|---|---|---|---|---|---|
| Model | 6482.46 | 9 | 720.27 | 11.81 | 0.0018 | Highly significant[a] |
| A | 809.60 | 1 | 809.60 | 13.27 | 0.0083 | Highly significant[a] |
| B | 3019.57 | 1 | 3019.57 | 49.50 | 0.0002 | Highly significant[a] |
| C | 1211.96 | 1 | 1211.96 | 19.87 | 0.0029 | Highly significant[a] |
| AB | 473.25 | 1 | 473.25 | 7.76 | 0.0271 | Significant[b] |
| AC | 35.03 | 1 | 35.03 | 0.57 | 0.4733 | Not significant[c] |
| BC | 425.97 | 1 | 425.97 | 6.98 | 0.0333 | Significant[b] |
| $A^2$ | 47.27 | 1 | 47.27 | 0.77 | 0.4079 | Not significant[c] |
| $B^2$ | 423.37 | 1 | 423.37 | 6.94 | 0.0337 | Not significant[c] |
| $C^2$ | 39.62 | 1 | 39.62 | 0.65 | 0.4468 | Not significant[c] |
| Residual | 427.00 | 7 | 61.00 | | | |
| Total deviation | 6909.46 | 16 | | | | |

[a]**Highly siginificant means that the P value is less than 0.01.**
[b]**Siginificant means that the P value is less than 0.05.**
[c]**Not siginificant means that the P value is greater than 0.05.**

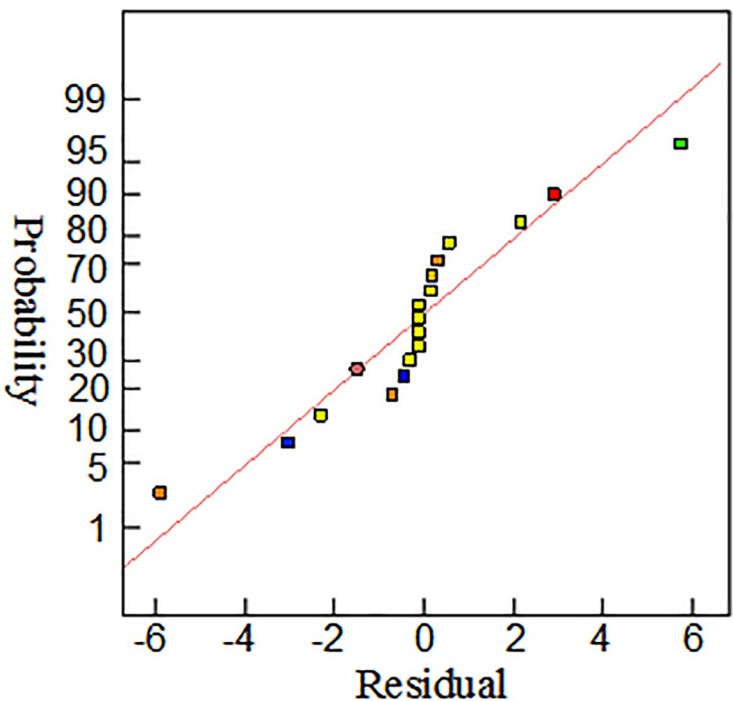

**Fig 10. Probability graph of residual normal distribution.**

equation. The residual error and probability distribution diagram of the equation and the probability distribution diagram of the predicted and actual values are shown in Figs 10 and 11, respectively. The scattered points are distributed around the residual error and probable line or best fitting line for the actual and predicted values, which indicate that the model of uniform crack propagation based on the response surface optimization method has good adaptability.

The comparison of the actual numerical simulation of the expansion of the 17 component multi-cluster fracturing model with the fracture propagation length predicted by the response regression Eq (15) obtained by the response surface optimization method is shown in Fig 12. It can be seen from Fig 12 that the predicted value of the response regression equation is in good agreement with the actual numerical simulated curve. Therefore, the accuracy of the response regression equation is proved once again, and the application of the response regression equation can be utilized to optimize the global range of fracture propagation and obtain the optimal perforation parameter combination.

## Response surface and contour plot

The response surface optimization method can overcome the shortfall that the orthogonal experiment can not give intuitive graphics, with the interaction between factors expressed by a three-dimensional response surface and a two-dimensional contour map. Combining the three-dimensional response surface and contour map enables the analysis of the effect of the interaction between the number of intermediate cluster perforations(A), the number of cluster perforations on both sides(B), and the diameter of intermediate cluster perforations(C), they are shown in Figs 13 and 14. Among them, the projection of the three-dimensional response

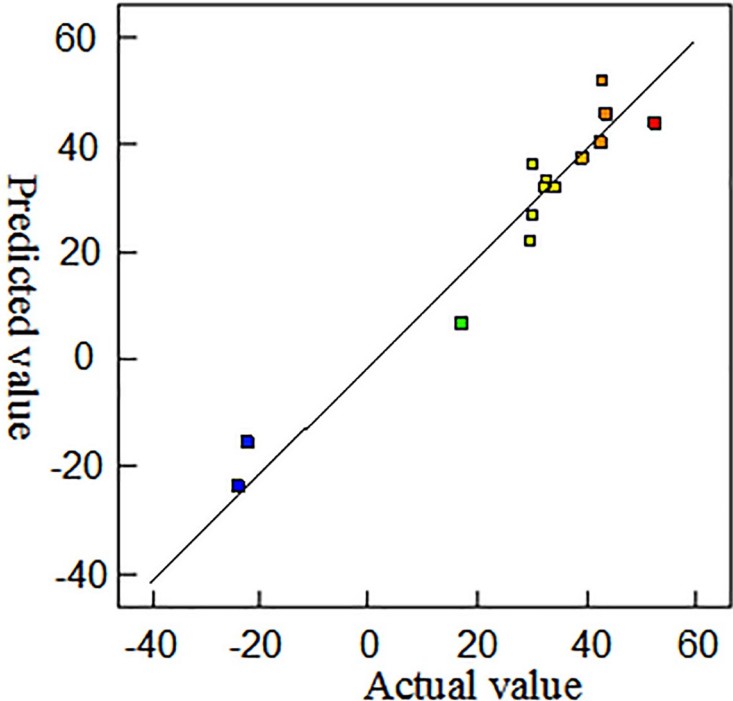

**Fig 11. Predicted value and actual distribution diagram.**

surface in the two-dimensional contour map is an ellipse that indicates that the interaction of the factors is significant, and a circle indicates that the interaction is not significant [32].

From the three-dimensional response graph and the two-dimensional contour map of Figs 13 and 14, it can be seen intuitively that the number of intermediate cluster perforations(A) and the number of cluster perforations on both sides(B) have a meaningful interaction. Moreover, within the range of 4–20, the seam length difference between clusters decreases as the number of perforating holes on both sides of the cluster decreased; As the number of intermediate cluster perforations increases, the seam length difference decreased.

From the three-dimensional response map and the two-dimensional contour map of Figs 15 and 16, it can be seen intuitively that the number of cluster perforations on both sides and the diameter of intermediate cluster perforations also have a significant interaction. In the range of 4–20 perforation holes, as the number of cluster perforations on both sides decreased, the seam length difference between clusters decreased. In the range of perforation diameter of 6-18mm, both the diameter of intermediate cluster perforations and the seam length difference are increased.

## Simulation verification

After analysing the relationship between the various factors and response surfaces, the optimal number of perforations and diameter of perforations predicted by the regression Eq (15) are as follows: the number of intermediate cluster perforations is 15, the number of cluster perforations on both sides is 7, and the diameter of intermediate cluster perforations is 15 mm. For the model with the best combination of parameters, simulation verification of segmented multi-cluster flow-limiting fracturing is carried out for which the cloud diagram of fracture propagation model before and after optimization is shown in Figs 17 and 18. It can be seen

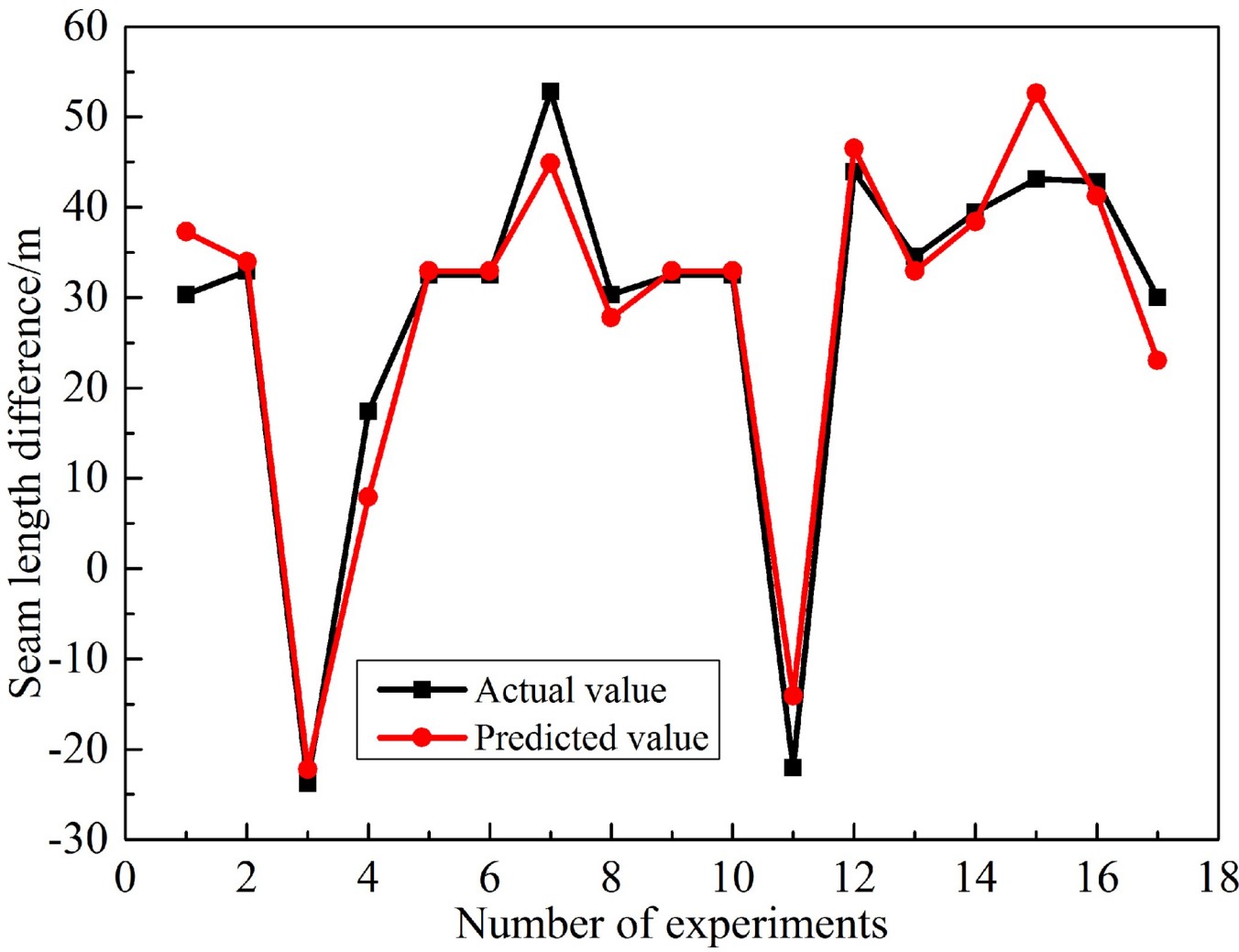

**Fig 12. The regression equation compares the predicted value with the actual value.**

clearly that before optimization in Fig 17, the fractures of the three cluster perforations developed on both sides while the intermediate cluster perforations do not develop. After optimization in Fig 18, the fractures of the three clusters become fully developed and no invalid clusters are formed. An evaluation table of perforation parameters before and after optimization is provided in Table 5. It is found that before optimization the seam length difference actual value is 32.550m, but after optimization the seam length difference actual value is 0.528m. From the contrast results of the seam length difference, it is found that the seam length difference between the perforation clusters before optimization and after optimization is greatly reduced. At the same time, in order to verify the accuracy of the response optimization equation, the predicted fracture propagation length between clusters before and after the optimization of perforation parameters was compared with the actual simulated value, Eq (16) is used to calculate the prediction error rate between the predicted value of the equation and the actual value, it is calculated that the error rate of the response equation for the judgment of the gap length before optimization is 1.2%, and the error rate for the judgment of the gap length after optimization is 0.4%. It is concluded that the response equation has a better predictive performance for the propagation of fractures, so the optimization parameters of the formation fracture

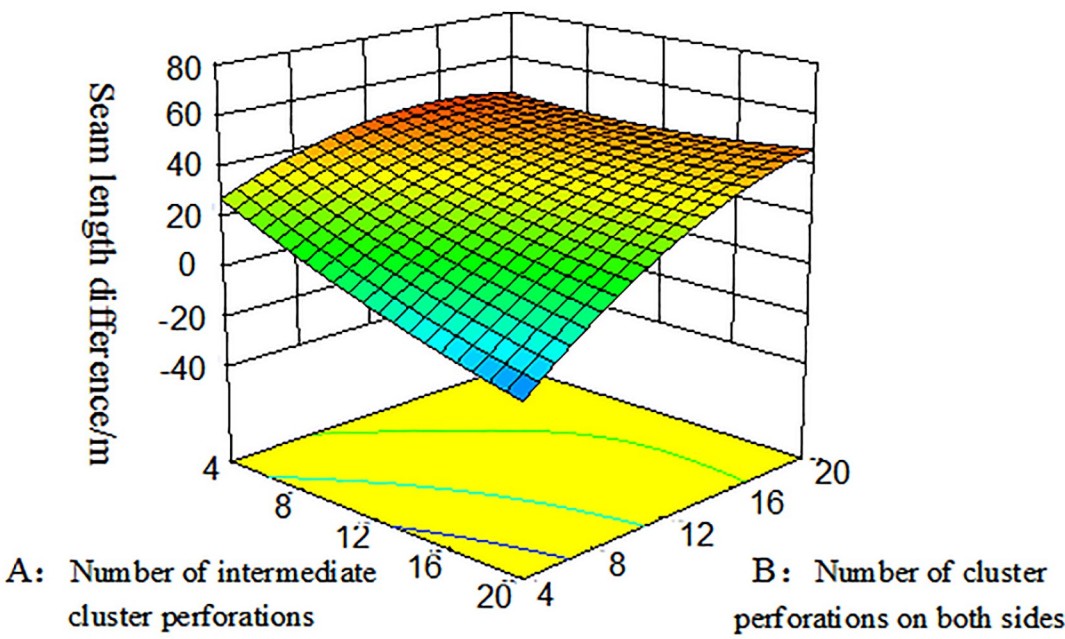

**Fig 13. Three-dimensional response surface diagram of the number of intermediate cluster perforations and the number of cluster perforations on both sides on the seam length difference.**

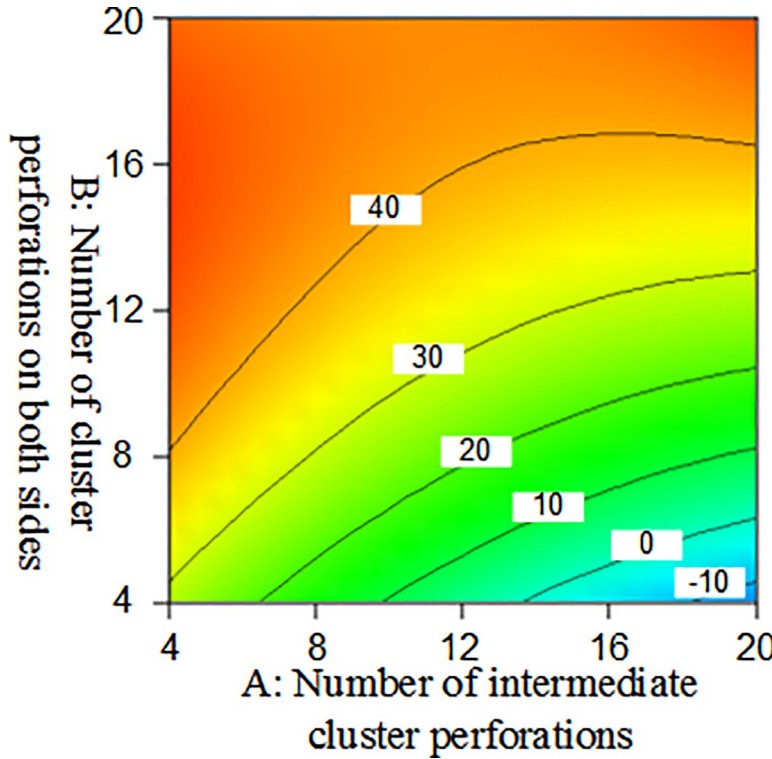

**Fig 14. Two-dimensional contour map of the number of intermediate cluster perforations and the number of cluster perforations on both sides on the seam length difference.**

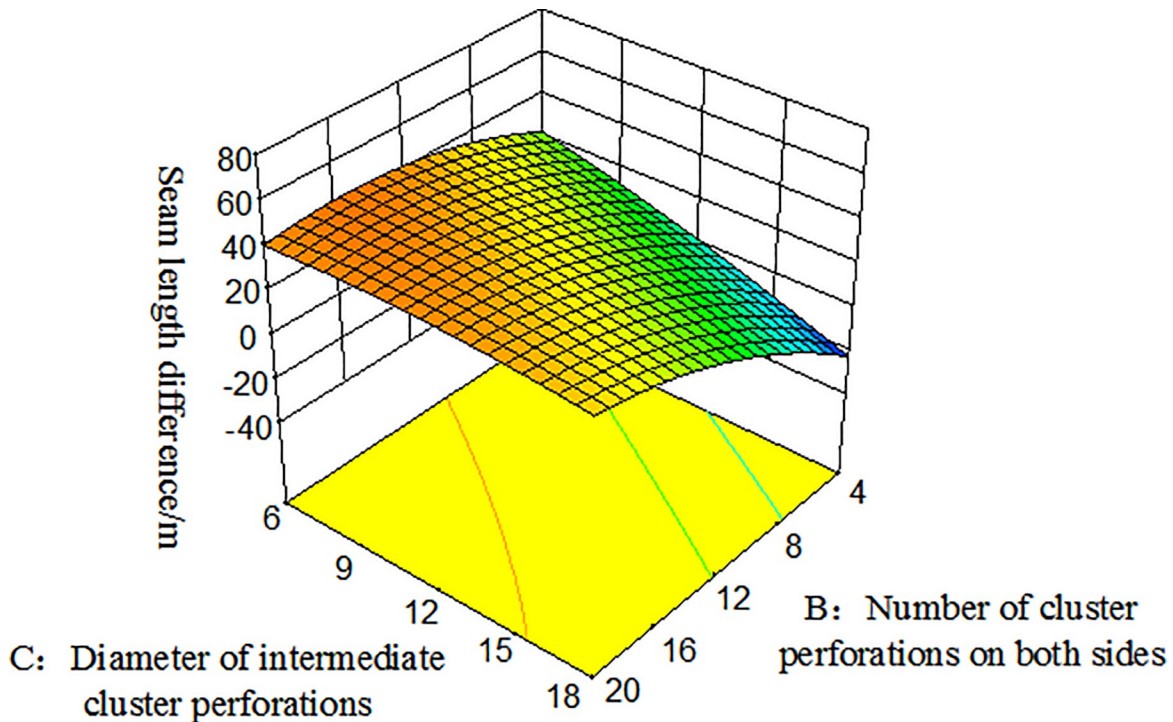

**Fig 15. Three-dimensional response surface diagram of the number of cluster perforations on both sides and the diameter of the middle cluster perforation hole on the seam length difference.**

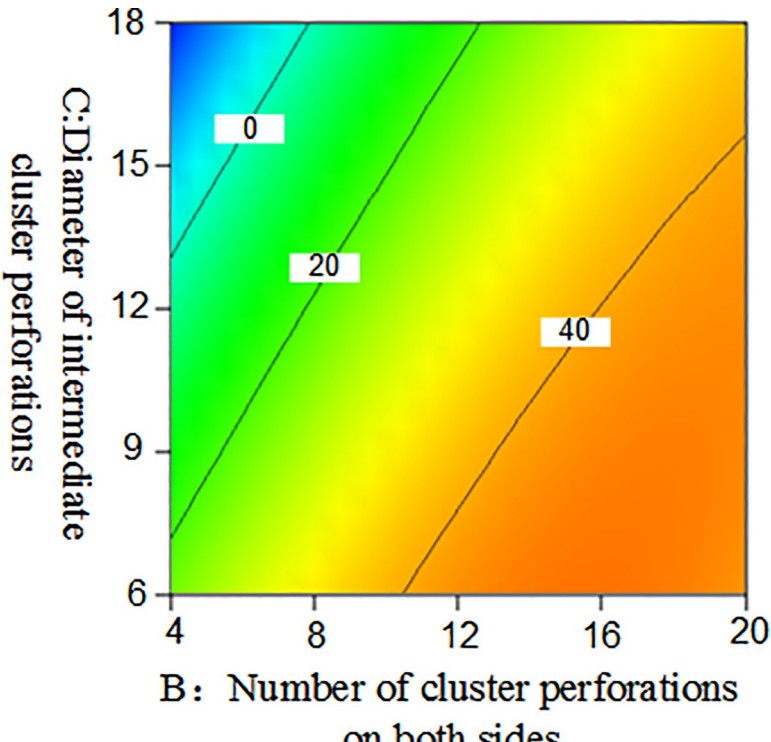

**Fig 16. Two-dimensional contour map of the number of cluster perforations on both sides and the diameter of the middle cluster perforation hole on the seam length difference.**

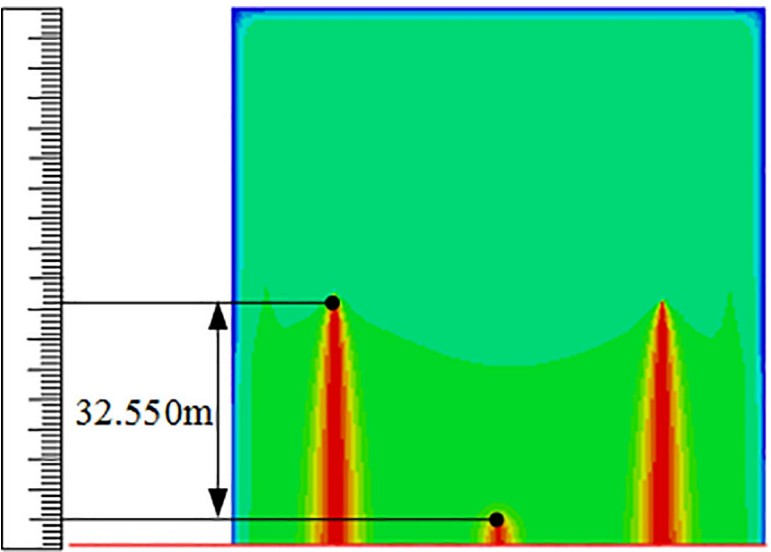

**Fig 17. Crack propagation shape before optimization.**

propagation obtained according to the equations have strong authenticity.

$$\boldsymbol{\varepsilon} = |(\boldsymbol{A} - \boldsymbol{E})/\boldsymbol{A}| \times 100\% \qquad (16)$$

Where $\boldsymbol{\varepsilon}$ represents estimated deviation rate, A represents actual value, E represents predictive value.

Finally, the quantity of flow distribution results in the three perforation clusters is extracted as shown in Figs 19 and 20. The injection flow rate of fracturing fluid at the wellhead is 0.03 m³/s. The quantity of flow distribution between clusters (Fig 19) before perforation cluster optimization shows that the sum of the flow rates of the perforation clusters on both sides is

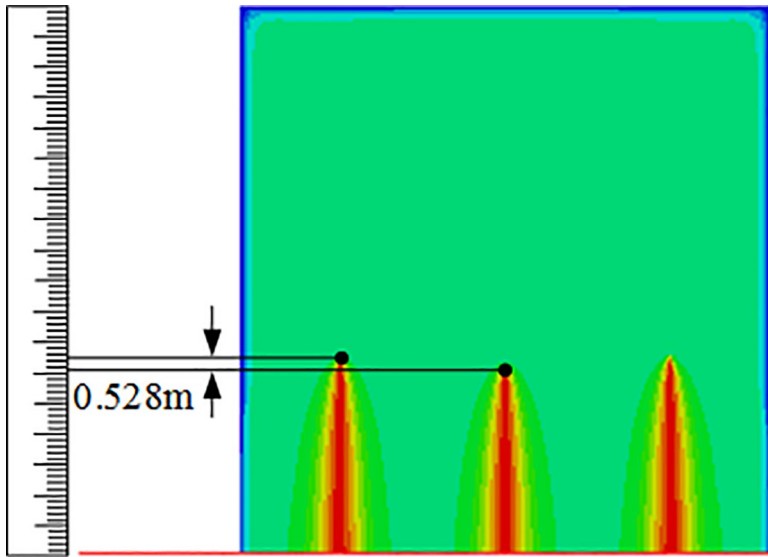

**Fig 18. Crack propagation shape after optimization.**

**Table 5. Evaluation table of perforation parameters before and after optimization.**

| Options | Number of intermediate cluster perforations(A) | Number of cluster perforations on both sides(B) | Diameter of intermediate cluster perforations (C)/ mm | Seam length difference actual value /m | Seam length difference predictive value/m | Estimated deviation rate (ε) |
|---|---|---|---|---|---|---|
| Before optimization | 12 | 12 | 12 | 32.55 | 32.95 | 1.20% |
| After optimization | 15 | 7 | 15 | 0.528 | 0.530 | 0.40% |

about 0.029 m$^3$/s. However, the intermediate perforation cluster flow distribution is only about 0.001 m$^3$/s. This is because the perforation parameters of each perforation cluster are the same before optimization, causing the results in the same perforation friction at each cluster perforation. Under the interference of inter-cluster stress, the intermediate cluster perforations quantity of flow distribution is very less, and the cracks hardly expand. After optimization of perforation clusters, the quantity of flow distribution among clusters (Fig 20) is as follows: the flow distribution among the three perforation clusters is approximately equal, which is about 0.01 m$^3$/s. The parameters of each cluster perforation after optimization are the best matching values under the interaction, which controls the difference in friction between each cluster

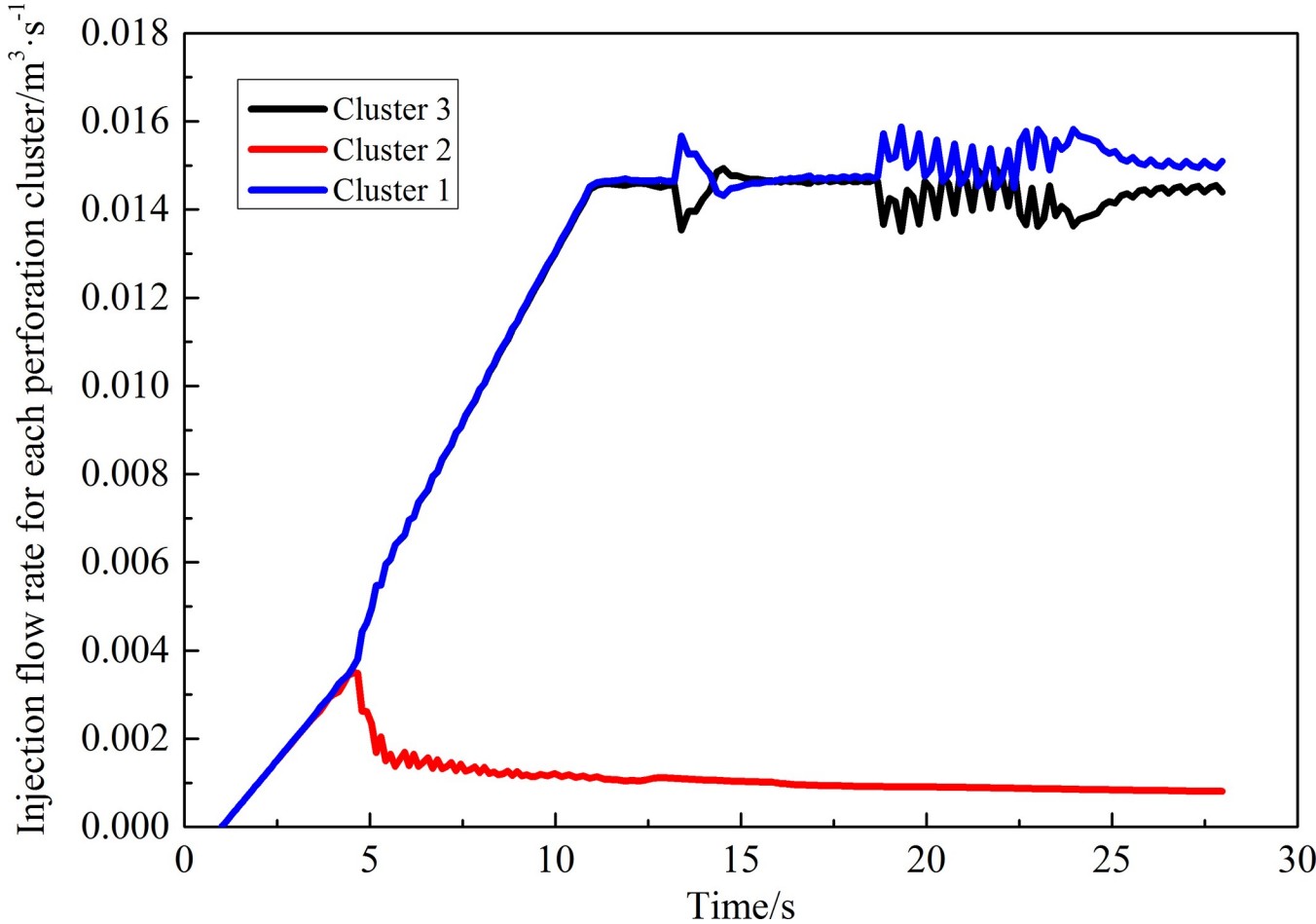

**Fig 19. Quantity of flow distribution between clusters before optimization.**

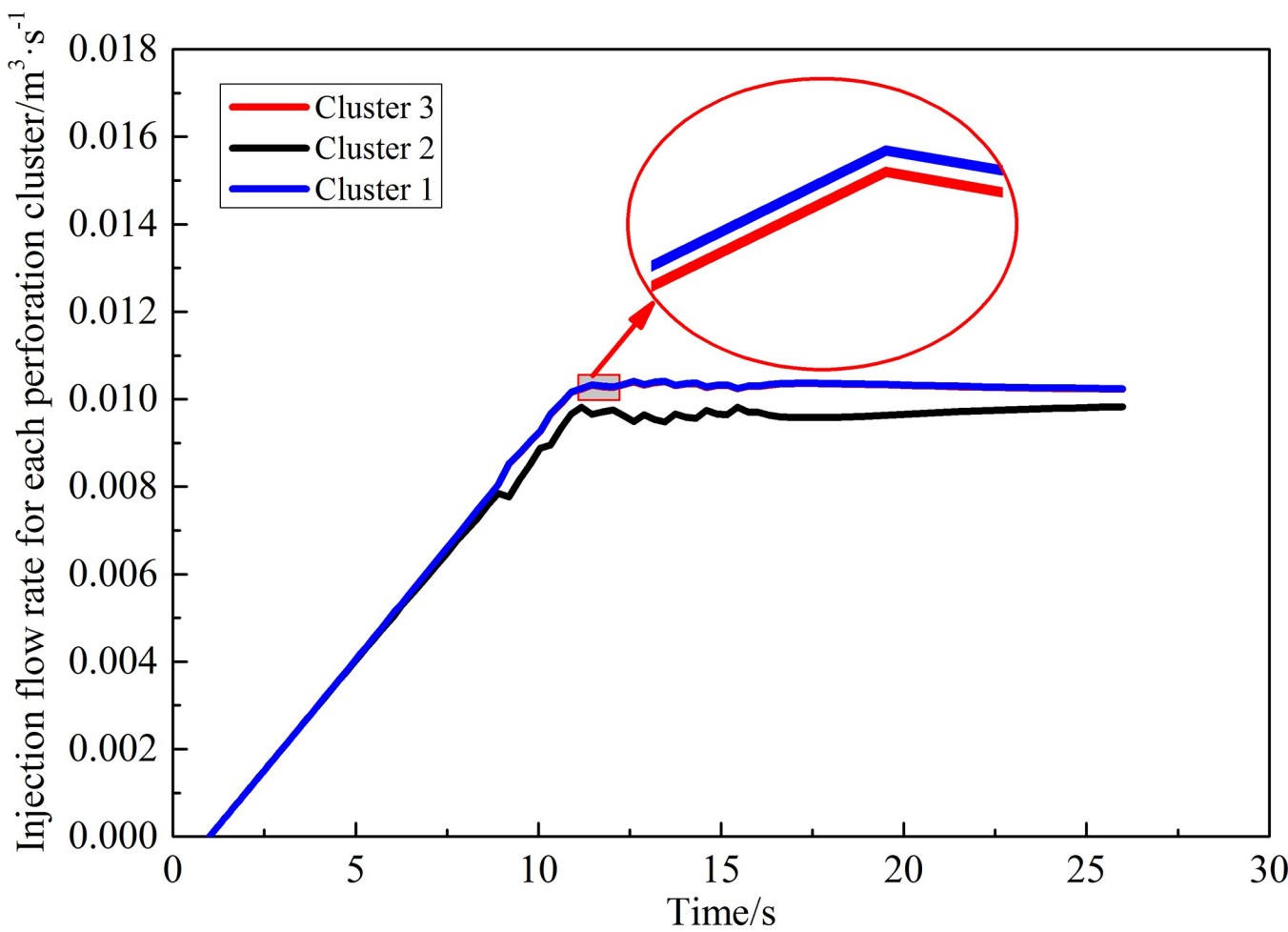

**Fig 20. Quantity of flow distribution between clusters after optimization.**

perforation and balances the flow distribution so that the cracks of each perforation cluster develop uniformly. It is proved that the perforation parameters after response optimization are more conducive to the initiation and propagation of segmented multi-cluster fracturing fractures.

## Conclusions

1. The newly established pipe flow and connection unit can control the difference of friction between perforations and realize the dynamic distribution of flow, which makes the numerical simulation process more consistent with the real dynamic fracturing process.

2. Based on the response surface optimization method, it is found that the number of intermediate cluster perforations, the number of cluster perforations on both sides, and the diameter of intermediate cluster perforations are all the most significant factors that affect the quantity of flow distribution on each cluster and lead to the different length of seam crack propagation. In the meanwhile, It is found that there is an obvious interaction between the number of intermediate cluster perforations and the number of cluster perforations on

both sides, the number of cluster perforations on both sides and the diameter of intermediate cluster perforations.

3. Based on the response regression equation obtained by the response surface optimization method, the global range can be quickly optimized and predicted, and the optimal perforation combination parameters are given. The deviation of the parameter estimation before and after optimization is verified. The predicted value and actual simulated value given by the equation is 1.2% and 0.4%. The estimated deviation rate shows that the equation has high accuracy.

## Author Contributions

**Resources:** Tiancai Cheng.

**Visualization:** Kangxing Dong.

**Writing – original draft:** Wei Liu.

**Writing – review & editing:** Suling Wang.

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
