## [Decision Letter · Decision Letter 0]

1 Jul 2021

PONE-D-21-19112

Research on optimization of perforation parameters for formation fractures based on response surface optimization method

PLOS ONE

Dear Dr. Liu,

Thank you for submitting your manuscript to PLOS ONE. After careful consideration, we feel that it has merit but does not fully meet PLOS ONE’s publication criteria as it currently stands. Therefore, we invite you to submit a revised version of the manuscript that addresses the points raised during the review process.

ACADEMIC EDITOR: The paper is well-written with publishable content. However, both reviewers notice that the manuscript writing needs to be further polished in terms of the quality of the presentation. In particular, typos need to be corrected, figures should be replaced by high-quality images. Some terminologies should have been used consistently, etc. Therefore, based on the current version, a major revision is necessary. 

We look forward to receiving your revised manuscript.

Kind regards,

Qichun Zhang, PhD

Academic Editor

PLOS ONE

Additional Editor Comments:

The paper is well-written with publishable content. However, both reviewers notice that the manuscript writing needs to be further polished in terms of the quality of the presentation. In particular, typos need to be corrected, figures should be replaced by high-quality images. Some terminologies should have been used consistently, etc. Therefore, based on the current version, a major revision is necessary.

Journal Requirements:

3. Please ensure that you refer to Figure 13 in your text as, if accepted, production will need this reference to link the reader to the figure.

Reviewers' comments:

Reviewer's Responses to Questions

**Comments to the Author**

1. Is the manuscript technically sound, and do the data support the conclusions?

Reviewer #1: Yes

Reviewer #2: Yes

2. Has the statistical analysis been performed appropriately and rigorously? 

Reviewer #1: Yes

Reviewer #2: Yes

3. Have the authors made all data underlying the findings in their manuscript fully available?

Reviewer #1: Yes

Reviewer #2: Yes

4. Is the manuscript presented in an intelligible fashion and written in standard English?

Reviewer #1: Yes

Reviewer #2: Yes

5. Review Comments to the Author

Reviewer #1: In this paper, the response surface optimization method is applied to the optimization of hydraulic fracturing perforation parameters, and the interaction between the parameters is considered. This paper has certain innovative significance and the value of guiding the subsequent engineering application. I have some of my questions and comments:

1. In Section " Unit and simulation method validation"

Figure 3(b) should mark cluster 1, cluster 2 and cluster 3. Otherwise, reader cannot be clearly recognized.

In Section "Calculation model"

The parameters of simulation calculation should be presented in tables, and the material parameters of simulation cannot be intuitively understood in the language of text description.

2. In Section " Calculation model"

Figure 5 should mark the setting part of pipe flow unit and connection unit. Otherwise, the relationship between pipe flow unit and connection unit mentioned in section "Material AND Methods" setting model is not clear.

In Section "Response optimization design method"

The response surface optimization method process in Figure 6 shows that the response surface update and response surface based optimization should be sequential, not parallel, Please amend.

3. In Section "Response surface optimization design scheme"

The differences between BBD and CCD methods as well as the central point were introduced. The central, cubic, and axial points of BBD and CCD should be described.

4. In Section "Model construction and test".

Why the equation based on response surface optimization can optimize the parameters under the global response. In what ways does the equation prove to be more predictive? Please specify.

Reviewer #2: Paper based on response surface optimization method completed the optimization of perforation parameters, the response surface method can give full consideration to the interactions between multiple factors, as well as for a global response under the mathematical model of forming a response equation, can achieve rapid optimization of perforation parameters, the paper has certain innovation.

Suggest some changes to the format, as follows:

1. In Section "Unit and simulation method validation"

There is little difference between the numerical simulation curve and the reference curve, so it is impossible to see at a glance whether the numerical simulation curve and the reference curve can be beautified to form a clearer expression. Please amend.

2. In Section " Simulation verification"

Figure12. Height difference between clusters is not clearly marked. Can the height difference between clusters be expressed in a better way? Please amend.

3. For the full text of the picture, the font size in the picture should be unified, there are some big and some small. Please amend.

4. The font of the full text table and the picture as well as the table should be unified. Some fonts in the picture and the table are not the same.Please amend.

5. In Section "Reference"

Some journal titles, such as Ref. 18, should be capitalized. Please amend.

6. PLOS authors have the option to publish the peer review history of their article (what does this mean?). If published, this will include your full peer review and any attached files.

Reviewer #1: No

Reviewer #2: No

---

## [Author Response · Author response to Decision Letter 0]

15 Jul 2021

Response to Editor：

Dear Editor, 

Thank you very much for your work on my paper.

I have further revised the paper according to the reviewers' suggestions.

Sincerely Yours,

Wei Liu

2021/07/09

Response to Reviewer #1:

Dear professor, 

Thank you for your comments to this paper. Now reply to your questions as follows.

Q1: In Section " Unit and simulation method validation"

Figure 3(b) should mark cluster 1, cluster 2 and cluster 3. Otherwise, reader cannot be clearly recognized.

A1: Figure 3(b) has been modified and cluster 1, cluster 2 , cluster 3 has been marked. Please see Figure 3(b) for details. Thank you.

Q2: In Section "Calculation model"

Figure 5 should mark the setting part of pipe flow unit and connection unit. Otherwise, the relationship between pipe flow unit and connection unit mentioned in section "Material AND Methods" setting model is not clear.

A2: Yes, you are right. If i can’t mark the setting part of pipe flow unit and connection unit, the reader may be not clear the relationship between pipe flow unit and connection unit mentioned. According to your opinion, Figure 5 has been modified. Please see Figure 5 for details.Thank you.

Q3: In Section "Response optimization design method"

The response surface optimization method process in Figure 6 shows that the response surface update and response surface based optimization should be sequential, not parallel, Please amend.

A3: Figure 6 has been modified. Please see Figure 6 for details.Thank you.

Q4: In Section "Response surface optimization design scheme"

The differences between BBD and CCD methods as well as the central point were introduced. The central, cubic, and axial points of BBD and CCD should be described.

A4: The author added a comparative description of BBD and CCD methods, and also added a different comparison diagram of BBD and CCD, which is shown in Figure 7 in the updated paper.Thank you .

Q5: In Section "Model construction and test"

Why the equation based on response surface optimization can optimize the parameters under the global response. In what ways does the equation prove to be more predictive? Please specify.

A5: First, response surface optimization (RSM) is an optimization method that integrates experimental design and mathematical modeling. By conducting tests on representative local points, the function relationship between factors and results in the global scope is regressed and fitted, and the optimal level value of each factor is obtained.The response surface optimization method in this paper is based on multiple linear regression equation, and 17 groups of test data are used for response regression analysis. Therefore, the optimization of perforation parameters in the global response can be completed. 

Second, to explain the equilibrium equation and have better prediction of perforation parameters,The value of the multivariate phase relation can reflect the accuracy of the fitting equation. If the correlation coefficient R-squared is approaching 1, it indicates that the response has a strong correlation. As a result of the experimental design scheme, the fitting correlation coefficient R-squared is 0.94, which is close to 1 in height, proving the accuracy of the fitting equation. At the same time, in table parameter changes before and after optimization, the prediction deviations of the fracture cluster length developed by the perforation parameters before and after optimization were calculated by using the response equation. The prediction deviations of the equation before and after optimization were 1.2% and 0.4%, respectively, indicating that the response optimization equation has good predictability.

Thank you for your comments to this paper again.

Sincerely Yours,

Wei Liu

2021/07/16

Response to Reviewer #2:

Dear professor, 

Thank you for your comments to this paper. Now reply to your questions as follows.

Q1: In Section "Unit and simulation method validation"

In Figure 4, there is little difference between the numerical simulation curve and the reference curve, so it is impossible to see at a glance whether the numerical simulation curve and the reference curve can be beautified to form a clearer expression. Please amend.

A1: Figure 4 has been modified. Please see Figure 4 for details.Thank you.

Q2: In Section " Simulation verification"

Figure12. Height difference between clusters is not clearly marked. Can the height difference between clusters be expressed in a better way? Please amend.

A2: Figure 12 has been modified. Please see Figure 12 for details.Thank you.

Q3: For the full text of the picture, the font size in the picture should be unified, there are some big and some small. Please amend.

A3: The text size of the pictures in the whole text has been adjusted uniformly. Please see all picture for details.Thank you.

Q4: The font of the full text table and the picture as well as the table should be unified. Some fonts in the picture and the table are not the same.Please amend.

A4: The tables and pictures in the full text have been modified. See the tables and pictures in the revised draft for details.Thank you.

Q5: In Section "Reference"

Some journal titles, such as Ref. 18, should be capitalized. Please amend.

A5: References have been modified as a whole, please refer to the references in the revised text for details.Thank you.

Thank you for your comments to this paper again.

Sincerely Yours,

Wei Liu

2021/07/16

---

## [Decision Letter · Decision Letter 1]

26 Jul 2021

Research on optimization of perforation parameters for formation fractures based on response surface optimization method

PONE-D-21-19112R1

Dear Dr. Liu,

We’re pleased to inform you that your manuscript has been judged scientifically suitable for publication and will be formally accepted for publication once it meets all outstanding technical requirements.

Kind regards,

Qichun Zhang, PhD

Academic Editor

PLOS ONE

Additional Editor Comments (optional):

After the major revision, the quality of the manuscript has been improved while all the comments on the previous version have been addressed. As the reviewers satisfy the current revised version, I recommend accepting this paper.

Reviewers' comments:

Reviewer's Responses to Questions

**Comments to the Author**

1. If the authors have adequately addressed your comments raised in a previous round of review and you feel that this manuscript is now acceptable for publication, you may indicate that here to bypass the “Comments to the Author” section, enter your conflict of interest statement in the “Confidential to Editor” section, and submit your "Accept" recommendation.

Reviewer #1: All comments have been addressed

Reviewer #2: All comments have been addressed

2. Is the manuscript technically sound, and do the data support the conclusions?

Reviewer #1: Yes

Reviewer #2: Yes

3. Has the statistical analysis been performed appropriately and rigorously? 

Reviewer #1: Yes

Reviewer #2: Yes

4. Have the authors made all data underlying the findings in their manuscript fully available?

Reviewer #1: Yes

Reviewer #2: Yes

5. Is the manuscript presented in an intelligible fashion and written in standard English?

Reviewer #1: Yes

Reviewer #2: Yes

6. Review Comments to the Author

Reviewer #1: After the authors' revision, the manuscript meets the requirements of the journal. I suggest to accept this manuscript.

Reviewer #2: The author revised this manuscript comprehensively according to the reviewers' comments, and it can be accepted at present form.

7. PLOS authors have the option to publish the peer review history of their article (what does this mean?). If published, this will include your full peer review and any attached files.

Reviewer #1: No

Reviewer #2: No

---

## [Editor Report · Acceptance letter]

29 Jul 2021

PONE-D-21-19112R1 

Research on optimization of perforation parameters for formation fractures based on response surface optimization method 

Dear Dr. Liu:

I'm pleased to inform you that your manuscript has been deemed suitable for publication in PLOS ONE. Congratulations! Your manuscript is now with our production department. 

Kind regards, 

on behalf of

Dr. Qichun Zhang 

Academic Editor

PLOS ONE